# Kotlin∇
## A shape-safe DSL for differentiable programming

**Breandan Considine**
McGill University

**Michalis Famelis**
Université de Montréal

**Liam Paull**
Université de Montréal

## Abstract

Kotlin is a statically-typed programming language with support for embedded domain-specific languages, asynchronous programming, and multi-platform compilation. In this work, we present an algebraically-based implementation of automatic differentiation (AD) with shape-safe tensor operations, written in pure Kotlin. Our approach differs from existing AD frameworks in that Kotlin∇ is the first shape-safe AD library fully compatible with the Java type system, requiring no metaprogramming, reflection or compiler intervention to use. A working prototype is available: `https://github.com/breandan/kotlingrad`.

## 1 Introduction

Many existing AD frameworks are implemented in dynamically-typed languages, like Python. Some frameworks are written in statically-typed languages, but only consider primitive data types, and do not attempt to verify the shape of multidimensional arrays. Those which do, either use dynamic type checking or relatively esoteric languages like Haskell (Piñeyro et al., 2019). In our work, we demonstrate a shape-safe AD framework which supports static type checking and inference on array programs in a widely-used programming language called Kotlin.

Differentiable programming has a rich history among dynamic languages like Python, Lua and JavaScript, with early implementations including projects like Theano (Bergstra et al., 2010), Torch (Collobert et al., 2002), and TensorFlow (Abadi et al., 2016). Similar ideas have arisen in statically-typed, functional languages, such as Haskell's Stalin∇ (Pearlmutter & Siskind, 2008b), DiffSharp in F# (Baydin et al., 2015) and recently Swift (Lattner & Wei, 2018). However, the majority of existing AD libraries have a loosely- or dynamically- typed DSL, and few support shape-safe array programming in a widely-adopted programming language. To our knowledge, Kotlin has no prior AD implementation. However, the language has several useful features for implementing a native AD framework. Kotlin∇ primarily relies on the following language features:

- **Operator overloading and infix functions** allow a concise notation for defining arithmetic operations on algebraic structures, i.e. groups, rings and fields. (Niculescu, 2011)
- **λ-functions** support functional programming, following Pearlmutter & Siskind (2008a,b); Siskind & Pearlmutter (2008); Elliott (2009, 2018), et al.
- **Extension functions** support extending classes with new fields and methods which can be exposed to external callers without requiring sub-classing or inheritance.

Kotlin∇ is an embedded domain-specific language (eDSL). Embedded programs may appear structurally and behave semantically unlike native code, but are syntactically valid by definition. eDSLs are often used to implement declarative languages, such as SQL/LINQ (Meijer et al., 2006), OptiML (Sujeeth et al., 2011) and other fluent interfaces (Fowler, 2005). With a sufficiently expressive host language, one can implement any other language as a library, without needing to write a lexer, parser, compiler or interpreter. With proper type constraints, users will receive code completion and static analysis from their favorite development tools, with no further effort required.

Submitted to Program Transformations for Machine Learning workshop at NeurIPS 2019, Vancouver, Canada.

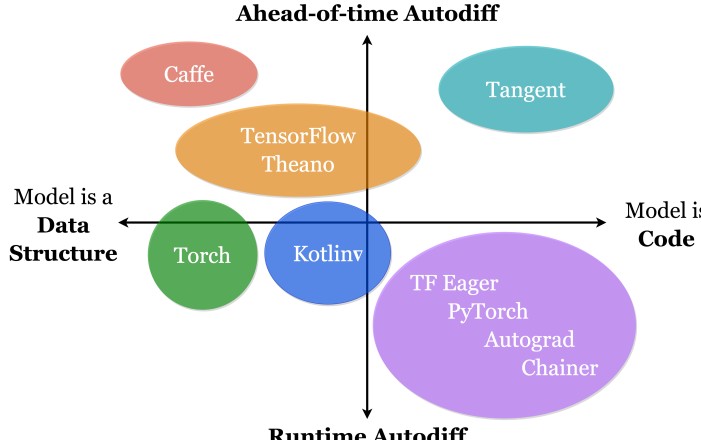

Figure 1: Adapted from van Merriënboer et al. (2018). Kotlin∇ models are data structures, constructed by an eDSL. These are compiled into dataflow graphs at runtime, which are eagerly optimized and lazily evaluated.

## 2 Usage

Kotlin∇ allows users to implement differentiable programs by composing simple functions to form more complex ones. Operations on functions with an incompatible output shape will fail to compile. Valid expressions are lazily evaluated inside a type-safe numerical context at runtime.

```
with(DoublePrecision) {                            // Use double-precision numerics
    val x = variable("x")                          // Declare immutable input variables
    val y = variable("y")                          // (these are just symbolic placeholders)
    val z = sin(10 * (x * x + pow(y, 2))) / 10     // Lazy expression
    val dz_dx = d(z) / d(x)                        // Leibniz derivative notation
    val d2z_dxdy = d(dz_dx) / d(y)                 // Mixing higher-order partials
    val d3z_d2xdy = grad(d2z_dxdy)[x]              // Gradient indexing operator
    plot3D(d3z_d2xdy, -1.0, 1.0)                   // Plot in -1 < x,y,z < 1
}
```

Figure 2: Above, we define a function with two variables and take a series of partial derivatives with respect to each variable. The function is evaluated on the interval $(-1, 1)$ in each dimension and rendered in 3-space.

$$z = \sin\left(10(x \times x + y^2)\right)/10, \quad \textbf{plot3D}\left(\frac{\partial^3 z}{\partial x^2 \partial y}\right)$$

Figure 3: Output generated by the program shown in Figure 2.

In Kotlin∇, all expressions are composed of function(s) in the host language which define a dataflow graph (DFG), and are themselves functions defined by the same DFG. An expression is only evaluated when invoked with numerical values. As shown in Figure 1, Kotlin∇ straddles the boundary between define-and-run and define-by-run. As an eDSL, it shares properties of both code and data.

## 3 Type System

Early work in type-safe dimension analysis can be found in Kennedy (1994, 1996) which uses types to encode dimensionality and prevent common bugs related to dimension mismatch from arising, and was later realized in the F# language (Kennedy, 2010). Jay & Sekanina (1997), Rittri (1995), and Zenger (1997) explore the application of dimension types for linear algebra. More recently, Kiselyov (2005); Kiselyov et al. (2009) and Griffioen (2015), show how to manipulate arrays in more complex ways. With the resurgence of interest in tensor algebra and array programming, Chen (2017) and Rink (2018) explore how to encode shape-safety in various type systems.

The problem we attempt to solve can be summarized as follows. Given two values `x` and `y`, and operator `$`, how do we determine whether the expression `z = x $ y` is valid, and if so, what is the result type of `z`? For matrix multiplication, when $x \in \mathbb{R}^{m \times n}$ and $y \in \mathbb{R}^{n \times p}$, the expression is well-typed and we can infer $z \in \mathbb{R}^{m \times p}$. More generally, we would like to infer the type of `z` for some operator $@ : (\mathbb{R}^{\mathbf{a}}, \mathbb{R}^{\mathbf{b}}) \rightarrow \mathbb{R}^{\mathbf{c}}$ where $\mathbf{a} \in \mathbb{N}^q, \mathbf{b} \in \mathbb{N}^r, \mathbf{c} \in \mathbb{N}^s$ and $q, r, s \in \mathbb{N}$. For many linear algebra operations such as matrix multiplication, $\mathcal{T}(\mathbf{a}, \mathbf{b}) \stackrel{?}{=} \mathbf{c}$ is computable in $\mathcal{O}(1)$ – we can simply check the inner dimensions for equivalence ($\mathbf{a}_1 \stackrel{?}{=} \mathbf{b}_0$).

```
val vecA = Vec(1.0, 2.0)          // Inferred type: Vec<Int, '2'>
val vecB = Vec(1.0, 2.0, 3.0)     // Inferred type: Vec<Int, '3'>
val vecC = vecB + vecB
val vecD = vecA + vecB // Compile error: Expected Vec<2>, found Vec<3>
```

Figure 4: Attempting to sum two vectors whose shapes do not match will fail to compile.

```
val matA = Mat('1', '4', 1.0, 2.0, 3.0, 4.0) // Inferred type: Mat<Double, '1', '4'>
val matB = Mat('4', '1', 1.0, 2.0, 3.0, 4.0) // Inferred type: Mat<Double, '4', '1'>
val matC = matA * matB
val matD = matA * matC          // Compile error: Expected Mat<4, *>, found Mat<1, 1>
```

Figure 5: Similarly, multiplying two matrices whose inner dimensions do not match will not compile.

Shape checking operations on multidimensional arrays is not always decidable. For arbitrary type functions $\mathcal{T}(\mathbf{a}, \mathbf{b})$, checking $\mathcal{T}(\mathbf{a}, \mathbf{b}) \stackrel{?}{=} \mathbf{c}$ requires a Turing machine. If $\mathcal{T}$ is allowed to use the multiplication operator, as in the case of convolutional arithmetic (Dumoulin & Visin, 2016), shape inference becomes equivalent to Peano arithmetic, which is undecidable (Gödel, 1931). Addition, subtraction, indexing and comparison of integers are all decidable operations (Charlier et al., 2011). Equality checking is trivially decidable, and can be implemented in most static type systems.

Evaluating an arbitrary $\mathcal{T}$ which uses multiplication or division (e.g. convolutional arithmetic) requires a dependently typed language (Xi & Pfenning, 1998; Piñeyro et al., 2019), but checking shape equality (e.g. ordinary arithmetic) is feasible in Java and its cousins.[1] Furthermore, we believe that shape checking ordinary matrix arithmetic is decidable in any type system loosely based on System $F_{<:}$ (Cardelli et al., 1994). We propose a type system for enforcing shape-safety which can be implemented in any language with subtyping and generics, such as Java (Naftalin & Wadler, 2007), Kotlin (Tate, 2013), TypeScript (Bierman et al., 2014) or Rust (Crozet et al., 2019).

## 4 Evaluation

Kotlin∇ claims to eliminate certain runtime errors, but how do we know the implementation is not incorrect? One method, called property-based testing (PBT) (Fink & Bishop, 1997), uses algebraic properties to verify the result of a calculation by constructing semantically equivalent but

---

[1]Java's type system is known to be Turing Complete (Grigore, 2017). Thus, emulation of dependent types in Java is theoretically possible, but likely intractable due to the practical limitations noted by Grigore.

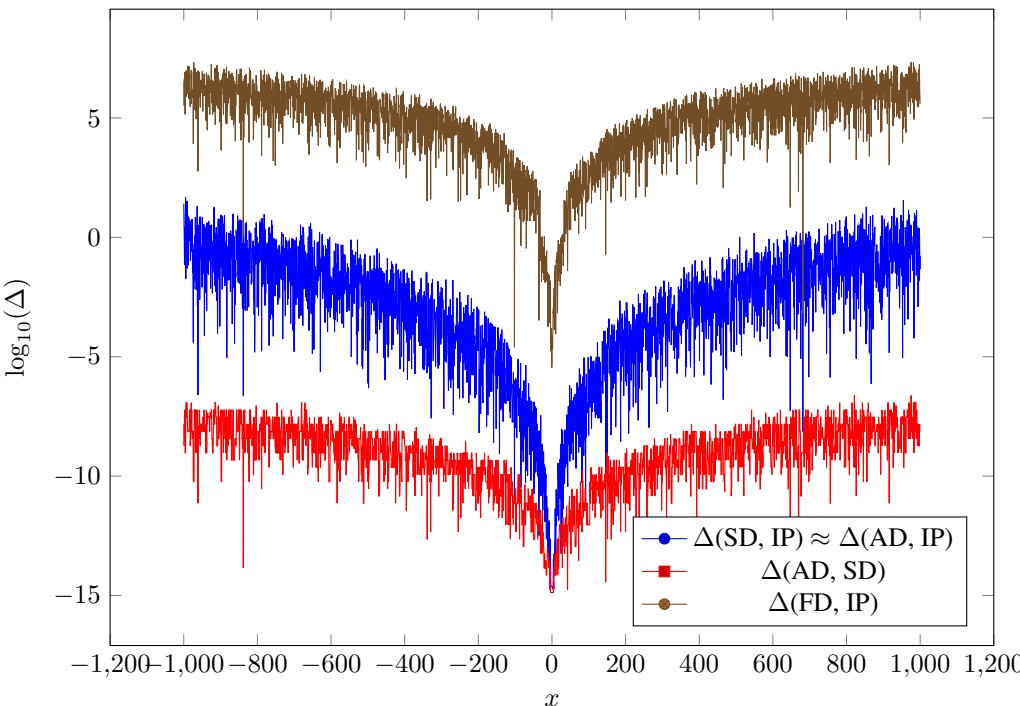

Figure 6: We compare numerical drift between three types of computational differentiation: (1) finite precision automatic differentiation (AD), (2) finite precision symbolic differentiation (SD) and (3) finite precision finite differences (FD), against infinite precision (IP) symbolic differentiation. AD and SD both exhibit relative errors (i.e. with respect to each other) several orders of magnitude below their absolute errors (i.e. with respect to IP), which roughly agree to within numerical precision. FD exhibits significantly higher drift than AD and SD.

syntactically distinct expressions. When evaluated on the same inputs, these should produce the same answer, to within numerical precision. Two such equivalences are used to to test Kotlin∇:

- **Analytical differentiation**: manually differentiate selected functions and compare the numerical result of evaluating random chosen inputs from their domain with the numerical result obtained by evaluating AD on the same inputs.
- **Finite difference approximation**: sample the space of symbolic differentiable functions, comparing the numerical results suggested by the finite difference method and the equivalent AD result, up to a fixed-precision approximation.

We also compare the precision of symbolic differentiation, automatic differentiation and numerical differentiation, as shown in Figure 6. These results are consistent with the findings of Laue (2019).

## 5 Conclusion

Unlike most existing AD implementations, Kotlin∇ does not require any template metaprogramming, compiler augmentation or runtime reflection to ensure type safety. Its implementation leverages several features in the Kotlin language including operator overloading, infix functions and extension functions. It also incorporates various functional programming concepts, like higher order functions, partial application and currying. The practical advantage of this approach is that it can be implemented as a simple library or embedded domain-specific language (eDSL), reusing the host language's type system to receive code completion and type checking for free. In future work, we hope to extend Kotlin∇ by compiling to an common intermediate representation (e.g. LLVM IR), and explore the meaning of differentiation in other calculi (cf. Considine (2019), Section 3.20).

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
