# OpenReview forum: "Kotlin∇: A shape-safe DSL for differentiable programming"
_NeurIPS.cc/2019/Workshop/Program_Transformations — Program Transformations @NeurIPS2019 Poster_

### Official Review · AnonReviewer2 · 2019-09-29
**Decent ideas but not fleshed out**

**Confidence:** 3
**Rating:** 6

**Review:**

This paper discusses a set of reasonable ideas and provides an interesting prototype demonstrating how the type system can be made more useful for numerical computing.  It seems like a bit of a mishmash of dimension types and autodiff, but the background is and related work discussion is thorough.

What's really missing is a bit more discussion of the limitations of this approach.

Questions:
 - what is meant by the "shape" of a return value?  For instance, what about lists and dicts of arrays?
 - I don't understand what is meant by "Kotlin∇ straddles the boundary between define-and-run and define-by-run"
 - In fig 6, how are the infinite precision methods producing numerical error?

---

> ### Public Comment · ~Breandan_Considine2 · 2019-10-06
> **Re: Decent ideas but not fleshed out**
>
> The authors wish to thank Anonymous Reviewer #2 for their constructive remarks in response to our submission.
>
> As the reviewer rightly noted, we omitted an important discussion about the limitations of the proposed approach, of which there are several. Primarily, Kotlin is not a dependently typed language, and we are forced to impose a fixed upper bound on the size of multidimensional arrays to statically check array programs. In practice, Kotlin’s type system is unable to statically shape-check programs with tensors whose size exceeds 65535 in any dimension, due to JVM constraints [1]. In addition, wall clock time required to statically type check array programs with shape values approaching this limit may be substantial, depending on the length of the program and the exact operations performed. We do not have precise benchmarks, but observed this issue becomes problematic for type values exceeding 1000 in the current compiler implementation (Kotlin 1.3.50).
>
> Indeed, Kotlin∇ enthusiastically borrows concepts from other programming languages and ADs. We concede, the concepts presented therein have been explored in prior literature, but their synthesis and integration in the Kotlin language are unique. Notably, our implementation of dimension types (with simplifying assumptions such as the one described above) demonstrates the feasibility of verifying shape properties in a blue-collar language without dependent types. Our implementation also inherits the classic disadvantages of static typing, such as the additional complexity of generic programming. We argue these costs are justified by the reduction in runtime errors.
>
> Re: What is meant by the “shape” of a return value?
>
> Let * be a matrix function, *: ℝ^(m×n), ℝ^(n×p) → ℝ^(m×p). We consider m×p to be the shape of the return value, which can be inferred from the shape of the function’s arguments (i.e. operands). In order to receive static type checking and inference for dynamically shaped structures (e.g. lists and dictionaries), users must explicitly declare the expected shape, otherwise type checking will only occur at runtime, prior to numerical evaluation. In both cases, shape is dynamically checked, but static analysis requires the author to either use array literals or explicitly declare the expected dimensionality of the underlying data structure. For array literals, Kotlin∇ uses source code generation to define a unique constructor for all array sizes below the fixed upper bound, allowing us to infer its dimension size based on the number of arguments supplied.
>
> Re: Define-and-run vs. define-by-run.
>
> As the reviewer correctly notes, this statement is vague and should have been clarified. In using it, we refer the nomenclature introduced by Tokui et al. (2015) [2], which roughly corresponds to lazy and eager execution, however, we find the latter terminology confusing. Unlike TensorFlow’s graph building mode [3], there is no explicit graph construction phase in Kotlin∇, although it does construct a similar dataflow graph with operator overloading, and performs compiler-like optimizations on that graph such as constant folding and propagation during runtime, but prior to numerical evaluation. Differentiation occurs on demand (i.e. only when a differential operator is encountered), and not during the primary graph construction phase. We claim it shares properties of both eager and lazy execution, although the boundary is somewhat blurry.
>
> Re: How are infinite precision methods producing numerical error?
>
> First, we note that any numerical evaluation of a trigonometric function on physical machinery will be approximate, although it was our intention to convey a slightly different phenomenon. In Figure 6, “arbitrary precision” is probably a more accurate choice of words, indicating a fixed, arbitrarily small precision relative to the other methods. To calculate IP, we symbolically derive the function in question and numerically evaluate it using arbitrary numerical precision (i.e. 30 significant figures) using the MacLaurin series expansion of sine and cosine. Figure 6 portrays the log difference between various forms of computational differentiation (evaluated using standard 32-bit floating point precision) and IP (computed to 30 significant figures). Please refer to the implementation [4] for further clarification, or do not hesitate to contact us if this is still unclear.
>
> We would again like to thank the reviewer for their thorough remarks and welcome further discussion.
>
> [1]: The Java Virtual Machine Specification: https://docs.oracle.com/javase/specs/jvms/se8/jvms8.pdf
> [2]: Chainer: a Next-Generation Open Source Framework for Deep Learning: http://learningsys.org/papers/LearningSys_2015_paper_33.pdf
> [3]: tf.Graph: https://www.tensorflow.org/api_docs/python/tf/Graph
> [4]: Comparison of AD, SD and FDM: https://github.com/breandan/kotlingrad/blob/9aefd6bd231841db0d34c6700e24b3b27681b4c2/src/main/kotlin/edu/umontreal/kotlingrad/samples/ADSDComparison.kt

---

### Official Review · AnonReviewer1 · 2019-09-30
**Interesting implementation**

**Confidence:** 4
**Rating:** 7

**Review:**

I like the type analysis, although of course ideally I’d like to see it in a non-staged-computation system.

The name sort of implies that the system does Reverse AD, to efficiently calculate gradients (“∇”) of high-dimensional systems. This isn’t clear from the description, which seems consistent with a system that only implements forward AD, and uses it to calculate gradients with O(n) overhead.

- The guts of Stalin∇ are described in Siskind and Pearlmutter (2008, Purdue TR-ECE-08-01, "Using Polyvariant Union-Free Flow Analysis to Compile a Higher-Order Functional-Programming Language with a First-Class Derivative Operator to Efficient {Fortran}-like Code", http://docs.lib.purdue.edu/ecetr/367). Also, Stalin∇ is in the Scheme family, not Haskell.

- I hate the use of the term “tensor” for arrays, but won’t belabour the point.

I think this probably falls in the poster category, because of its scope etc.

QUESTION FOR AUTHOR: Does Kotlin∇ support Reverse AD?

---

> ### Public Comment · ~Breandan_Considine2 · 2019-10-28
> **Re: Interesting implementation**
>
> We are grateful for the insightful comments raised by Reviewer #1 and apologize for the delayed response.
>
> Re: Stalin∇ is in the Scheme family
>
> The authors regret this conspicuous error and appreciate the provided reference material.
>
> Re: “Tensor” vs. “Multidimensional array”
>
> We acknowledge the term “tensor” is a contentious one, but have adopted it with precedent [1, 2] for idiomatic reasons, not mathematical correctness. A more precise term would be “multidimensional array”. We have no objection to the latter, but prefer the former for brevity.
>
> Re: Does Kotlin∇ support Reverse AD?
>
> In short, Kotlin∇ allows both forward and reverse accumulation. For functions ℝᵐ→ℝⁿ, forward mode is preferred when m << n and reverse mode when n << m.
>
> More specifically, Kotlin∇ implements symbolic differentiation (SD), of which we consider forward and reverse mode AD to be special cases, corresponding to left- and right- associativity of the matrix multiplication operator. It is our understanding that "automatic differentiation" is a somewhat controversial term in the AD literature and have tried to use the term sparingly. We believe this discussion deserves a far more detailed treatment, but would like to call the reader’s attention to some recent work which has lead us to this conclusion.
>
> The distinction between AD and SD becomes increasingly blurry when considering flexible execution models and hybrid ADs capable of both eager [4] and lazy [5] execution, which in our view, are both automatic differentiation. We believe symbolic differentiation is a form of differentiation which the AD literature has been too quick to dismiss. SD in particular, gives the compiler far more flexibility to perform global optimizations such as algebraic simplification [6], loop vectorization [7] and tensor comprehensions [8], which are unclear how to implement using forward- or reverse-mode accumulation in its direct form.
>
> Again, we thank Reviewer #1 for their insightful remarks and are open to changing our view in light of further discussion.
>
> [1] Chen, T. Typesafe abstractions for tensor operations. http://doi.acm.org/10.1145/3136000.3136001
> [2] Rink, N. Modeling of languages for tensor manipulation. http://arxiv.org/pdf/1801.08771.pdf
> [3] Wang, F., et al. Demystifying differentiable programming: Shift/reset the penultimate backpropagator. https://arxiv.org/pdf/1803.10228.pdf
> [4] Agrawal, A., et al. TensorFlow Eager: A multi-stage, Python-embedded DSL for machine learning. https://www.sysml.cc/doc/2019/88.pdf
> [5] Looks, M., et al. Deep learning with dynamic computation graphs. https://arxiv.org/pdf/1702.02181.pdf
> [6] Bergstra, J., et al. Theano: a CPU and GPU math expression compiler. http://conference.scipy.org/proceedings/scipy2010/pdfs/bergstra.pdf
> [7] Agarwal, A. Static automatic batching in TensorFlow. http://proceedings.mlr.press/v97/agarwal19a/agarwal19a.pdf
> [8] Vasilache, N., et al. Tensor Comprehensions: Framework-agnostic high-performance machine learning abstractions. https://arxiv.org/pdf/1802.04730.pdf

---

### Public Comment · ~Andreas_Griewank2 · 2019-10-02
**Nothing Much New**

We have seen something like Figure 6 hundreds of times. But there is still confusion about the difference between
"automatic", "symbolic" and "analytic" differentiation.  In my book they are all the same, of course given
the same arithmetic accuracy. Anyhow, I prefer to call the stuff "algorithmic" differentiation. Cannot really judge the "typing" aspect.

---

> ### Public Comment · ~Breandan_Considine2 · 2019-10-06
> **Re: Nothing Much New**
>
> The authors wish to thank Prof. Griewank for reviewing our ongoing work. A well-worn copy of “Evaluating Derivatives: Principles and Techniques of Algorithmic Differentiation” [1] sits on our desk, and we share his view that symbolic differentiation (SD) and automatic/algorithmic differentiation (AD) are indeed the same creatures. In our view, forward and reverse mode AD are special cases of symbolic differentiation, corresponding to left- and right- associativity of the chain rule, as many others, including the authors of [1] have previously shown. However, this view appears somewhat controversial in the AD community, and Figure 6 is our small attempt to reexamine this claim from an empirical perspective. We simply use it to demonstrate our approach to testing, which compares algorithmically distinct but semantically equivalent differentiation algorithms, and make no claim regarding its novelty. But we also hope to provoke further discussion on what, if any, are the differences between SD and AD, a distinction we have struggled to appreciate.
>
> Others in the AD community have insisted “AD is not SD” [2], although this claim has recently been called into question, cf. “AD *is* SD performed by a compiler,” (Elliott [4]), and “AD and SD are equivalent in the sense that they both perform the same operations,” (Laue [5]). We realize the AD/SD discussion deserves a far more nuanced treatment that we can provide in this rebuttal, but would like to understand the arguments more clearly. Why is SD considered inferior to AD, when they appear to perform the same calculations? Why, as suggested by prior literature [2], is the problem of “expression swell” unique to SD? As shown by Laue [5], whose findings we can numerically reproduce, “expression swell” does not appear to harm SD, or confer any advantage to AD. While certain ADs may interleave numerical evaluation and differentiation, is that execution model such an important departure from SD? By performing separate passes over the dataflow graph for differentiation and numerical evaluation, it seems that SD is able to perform far more powerful algebraic simplifications (cf. Gunter [6]) than AD as presented by Baydin [2], which is locked to the primal dataflow graph in its textbook form. What advantage does AD have, if any, by performing evaluation and differentiation in lockstep?
>
> Again, we wish to thank Prof. Griewank for his valuable remarks and would be grateful for any further insight he (or others) can provide to help reconcile these two -- seemingly contradictory -- views in the AD/SD literature.
>
> [1] Griewank, A and Walther, A. “Evaluating Derivatives: Principles and Techniques of Algorithmic Differentiation” https://books.google.ca/books?id=xoiiLaRxcbEC
> [2] Baydin, et al., “Automatic Differentiation in Machine Learning: a Survey” www.jmlr.org/papers/volume18/17-468/17-468.pdf
> [4] Elliott, C. “The Simple Essence of Automatic Differentiation” https://arxiv.org/pdf/1804.00746.pdf
> [5] Laue, S. “On the Equivalence of Forward Mode Automatic Differentiation and Symbolic Differentiation” https://arxiv.org/pdf/1904.02990.pdf
> [6] Guenter, B. “Efficient Symbolic Differentiation for Graphics Applications” https://www.microsoft.com/en-us/research/wp-content/uploads/2016/02/main-65.pdf

---

### Decision · Program_Chairs · 2019-10-01

**Decision:**

Accept (Poster)

**Comment:**

The reviewers agreed that this is a good contribution to the workshop but that its scope makes it more appropriate for a poster presentation.

---

> ### Public Comment · ~Breandan_Considine2 · 2019-10-28
> **Re: Paper Decision**
>
> The authors wish to express their appreciation to the PTML program chairs for carefully considering our work. We are delighted to receive their decision and are looking forwarding to participating in the workshop.